# Antibody Binding Captures High Energy State of an Antigen: The Case of Nsp1 SARS-CoV-2 as Revealed by Hydrogen–Deuterium Exchange Mass Spectrometry

**DOI:** 10.3390/ijms242417342

**Published:** 2023-12-11

**Authors:** Ravi Kant, Nawneet Mishra, Michael L. Gross

**Affiliations:** 1Department of Chemistry, Washington University, St. Louis, MO 63130, USA; ravi.kant@wustl.edu; 2Department of Pathology and Immunology, Washington University School of Medicine in St. Louis, St. Louis, MO 63110, USA; nmishra@wustl.edu

**Keywords:** Nsp1, epitope, paratope, hydrogen–deuterium exchange, mass spectrometry, antibody capture of high energy antigen

## Abstract

We describe an investigation using structural mass spectrometry (MS) of the impact of two antibodies, 15497 and 15498, binding the highly flexible SARS-CoV-2 Nsp1 protein. We determined the epitopes and paratopes involved in the antibody–protein interactions by using hydrogen–deuterium exchange MS (HDX-MS). Notably, the Fab (Fragment antigen binding) for antibody 15498 captured a high energy form of the antigen exhibiting significant conformational changes that added flexibility over most of the Nsp1 protein. The Fab for antibody 15497, however, showed usual antigen binding behavior, revealing local changes presumably including the binding site. These findings illustrate an unusual antibody effect on an antigen and are consistent with the dynamic nature of the Nsp1 protein. Our studies suggest that this interaction capitalizes on the high flexibility of Nsp1 to undergo conformational change and be trapped in a higher energy state by binding with a specific antibody.

## 1. Introduction

The onset of the SARS-CoV-2 pandemic in 2019 underscores the importance of expeditiously developing reagents to comprehend better antigen properties, viral pathogenesis, and host reactions and to facilitate diagnostic advances. To add more tools targeting SARS-CoV-2, previously, Amarasinghe and coworkers expressed and purified 21 recombinant SARS-CoV-2 proteins, carried out antibody (Ab) selections via phage display, and validated binding of the IgGs in vitro by performing Ab selections using phage display techniques [1] These mAbs were assessed for activity in Western blot (WB) and immunofluorescence (IF) assays utilizing SARS-CoV-2-infected cells. Although these findings suggest that these synthetic antibodies may be used in the investigation of SARS-CoV-2 viral proteins and for the development of novel diagnostic assays for COVID-19, the findings also invite questions about the location of binding and the effect of the highly flexible nonstructural protein (Nsps) antigens on antibody binding.

The Nsps of SARS-CoV-2 are remarkably flexible. Among these proteins, Nsp1 stands out owing to its ability to satisfy many functions in betacoronaviruses (b-CoVs), including SARS-CoV-2. Nsp1 can inhibit cellular mRNA translation, redirect the translational machinery to viral RNA templates, induce cell cycle arrest in the G0/G1 phase, and degrade cellular messenger RNAs [2,3]. Additionally, Nsp1 proteins may play a vital role in the innate immune response, downregulating the expression of virus-specific genes and shutting down host translation. The multifunctional nature of Nsp1 requires it to exhibit structural flexibility to interact effectively with various viral and host factors, enabling it to carry out efficiently its diverse functions. Therefore, understanding the structural flexibility of Nsps is highly important for deciphering their mechanisms of action and devising targeted interventions against SARS-CoV-2.

Typically, the interaction of an antibody and an antigen leads to the stabilization of the antigen. Although “localized destabilization” was reported in some instances [4], it is uncommon. Thus, antibody binding that reports on the flexibility of Nsp1 would be of high interest. In this study, we characterized the binding of two antibodies (Fabs 15497 and 15498) with Nsp1 by using hydrogen-deuterium exchange mass spectrometry (HDX-MS).

The standard bottom-up HDX-MS process involves these step [5,6,7,8,9,10]: (1) incubation with D_2_O (labeling), (2) quenching of the HDX by adding acidified denaturant, (3) digestion of the Nsp1 and/or antibody with acid-stable proteases, (4) chromatographic separation of peptides followed by mass spectrometric measurement of their isotope clusters, and (5) semiautomatic data analysis. Proteins in the solution are mixed into a D_2_O-based buffer in the first step, allowing labile hydrogens to exchange with deuterium from the solvent. Both backbone (-CON−H) and some side-chain hydrogens exchange, but HDX of protein side chains is not measured owing to their rapid exchange in and out following the quench. Amide hydrogens in the backbone, however, exchange slowly (minutes to hours), allowing them to be detected with MS. HDX is catalyzed by both acid and base, giving rise to a minimum exchange rate occurring at pH around 2.5, motivating the choice of an acid quench. Using a combination of proteases ensures the generation of small- to medium-sized peptides over most of the protein, offering detailed, good spatial resolution. Semi-automated data analysis software such as HDExaminer 2.5.1 calculates the mass shift due to deuterium incorporation. The software examines the isotopic distribution of a peptide or protein in its undeuterated state and establishes its centroid. Comparing this with the shifted centroid at later time points reveals the most probable extent of HDX. The results of the analysis are pictured as a kinetics graph plotting deuterium uptake (Y-axis) against time (X-axis). In most HDX experiments, researchers conduct differential footprinting, comparing two states (e.g., free versus ligand-bound or wild type versus mutant). Deuterium differences are computed by subtracting HDX values between these states, uncovering both local and allosteric impacts of the binding.

Our previous HDX findings regarding Nsp1 show that approximately one-third of the protein lacks a compact, well-defined structure consistent with its flexibility and posing challenges for crystallizing the full-length Nsp1. This observation aligns with the crystal structure of Nsp1 [11,12] , which had to be taken of a truncated version spanning residues from 10 to 127. Notably, we chose HDX for this study because HDX-MS is rapid, solution-based, sensitive to protein dynamics and flexibility and does not depend on protein size or complexity.

The question we ask is whether antibody binding can capture Nsp1 structural dynamics. We report here that Nsp1 binding traps a flexible, high-energy state of Nsp1, a phenomenon that may play a role in the development of therapeutic strategies targeting flexible proteins. Added flexibility increases entropy and would be expected to adversely diminish the ability of a protein to carry out its many functions.

## 2. Results

### 2.1. Fabs Bind to Distinct Domains on Nsp1

To determine the epitopes of antibodies 15497 and 15498 and assess changes in flexibility after antibody binding, we opted for HDX-MS (described above) to determine epitopes and large-scale remote conformational changes [4,5] . We conducted HDX of full-length Nsp1 with and without antibodies at ~4 °C. We opted for lower temperatures to compensate in part for the protein’s inherent flexibility, intentionally decelerating the hydrogen–deuterium exchange (HDX) process to enhance its sensitivity to temporal variations. To obtain comprehensive protein coverage, we utilized a combination of in-solution Fungal XIII protease and on-line immobilized pepsin to digest Nsp1. This approach generated a total of 101 unique peptides, with 100% coverage of its sequence and an average peptide length of 13 amino acids (Appendix A). The epitope-determining HDX experiments are differential experiments, where the deuterium uptake of the free antigen and antibody-bound antigen were compared by subtracting the deuterium uptake for the bound state from that of the unbound state.

The epitope mapping of Nsp1 with antibody 15497 revealed that the majority of the Nsp1 structure exhibited no significant difference in HDX between the bound and unbound states and hence no significant changes in high-order structure. An exception is the specific segment 150–157 that exhibited a decrease in HDX for the bound state, suggesting that the region contains the epitope (Figure 1A,C). Previous studies confirm that this region contains a small helix (residues 154-160) when it is bound to the 40S ribosome [13,14] When antibody 15497 is not bound, the peptides comprising this segment exhibit a rapid deuterium uptake, suggesting that there is no stable secondary structure in that region.

The observation of binding at this site is consistent with our recent HDX study on full-length Nsp1, in which we reported that this short helix is absent in the native protein [15] . Thus, the antibody 15497 binds specifically to the unstructured segment located toward the C terminus of Nsp1. Regrettably, due to the absence of the C-terminus in the crystal structure (that part of the protein was truncated), we are unable to depict the epitope of 15497 on the crystal structure. Plots of the HDX kinetics for the peptides illustrate a more pronounced difference in HDX at earlier time points, whereas at later time points, no significant differences are observed between the bound and unbound states (Appendix A). This pattern can be attributed to a binding event characterized by large on and off rates. The epitope findings are also in agreement with the Western blot analysis [1] that showed a loss of binding between antibody 15497 and Nsp1 when residues 128–180 were truncated.

Epitope mapping of antibody 15498 on Nsp1 reveals a different pattern of HDX. In the bound state, peptides covering 20-27 and 71-85 show a decrease in HDX in two distinct segments of the N-terminal domain (NTD) of Nsp1 (Figure 1B,C). The crystal structure shows that the first segment corresponds to a loop followed by a helix, whereas the second segment encompasses a loop between C terminal end of the β4 strand and the N-terminal portion of a β4 strand [12] (Figure 1D). These regions are entirely consistent with antigen binding and confirmed by the decrease in HDX at those sites. The peptides corresponding to the first segment exhibit differences in HDX, primarily at extended time points. This can be attributed to the length of the peptides, as localized effects might be diluted by neighboring residues that are not directly engaged in the binding process. These findings agree with the Western blot results [1] that support the N-terminal domain (NTD) of Nsp1 (residues 1-127) as a potential binding site for antibody 15498.

Overall, epitope mapping confirms that antibody 15497 binds to a linear epitope in the C-terminus, whereas 15498 targets a conformational epitope in the N-terminus of Nsp1. More importantly, the loss of the high-order structure over Nsp1 upon binding to 15498 is most unusual, and we hypothesize that the increase in HDX over the whole protein indicated that 15498 captured a high-energy state of Nsp1 in which the protein became less structured.

### 2.2. Antibody Binding Causes Widespread Conformational Change

Unlike the binding of Fab 15497 and Nsp1, which involves only the C-terminus of Nsp1, the binding of Fab 15498 to Nsp1 is surprisingly accompanied by significant increases in HDX across nearly all of the structural elements of the Nsp1-NTD (Figure 1B). It is intriguing that the kinetic plots depicting Nsp1 interactions, with and without antibodies, revealed that despite global delocalization, 15497 maintained its binding to Nsp1, indicating a strong and resilient interaction. This unexpected result suggests that the binding of Fab 15498 either triggers a widespread opening of the conformation of Nsp1 or captures a high-energy state of Nsp1. The resulting complex of 15498 contains an even more flexible state of the antigen where the secondary structure has been destabilized over the entire antigen-terminal domain.

### 2.3. HDX-MS Identifies the CDRs

To confirm the binding of the antigen to the two antibodies, our next focus was the paratopes (complementarity-determining regions or CDRs) on the Fabs that are involved in the interaction with Nsp1. Similar to epitope mapping experiments, we employed Fungal XIII solution along with immobilized pepsin to generate a series of overlapping peptides. The heavy chain of 15497 yielded 152 peptides, each with an average length of 14. The light chain of 15497 produced 113 peptides, with an average length of 13 (Appendix A). Notably, we achieved a sequence coverage of 100% for the heavy chain and 99% for the light chain. In the case of 15498, the heavy chain generated 98 peptides, averaging 15 residues each. The light chain of 15498 resulted in 65 peptides, averaging 13 residues each (Appendix A). We obtained full sequence coverage for the heavy chain and 98% for the light chain. The HDX data collection and analysis were similar to that for epitope mapping, except Nsp1 was added in excess to the Fab in a 1:2 ratio (Nsp1: Fab) to ensure binding.

The HDX analysis was performed separately for the light and heavy chain sequences of the Fabs. Peptides representing the CDRs H2 and H3 of the heavy chain and CDRs L2 and L3 of the light chain of antibody 15497 exhibit a decrease in HDX in the bound state vs. the unbound state, indicating involvement in the interactions with Nsp1 (Figure 2A,B). Similarly, for antibody 15498, the CDRs H1 and H3 of the heavy chain and CDRs L2 and L3 of the light chain are involved in the interaction with Nsp1 (Figure 2C,D). This is strong evidence that the antibodies are indeed binding to the Nsp1.

### 2.4. SEC MALS Shows Differences between 15497 and 15498

Before initiating the HDX experiments, we employed size exclusion chromatography coupled with multi-angle light scattering (SEC-MALS) to investigate the formation of the Fab-Nsp1 complex. The Fab-Nsp1 complexes eluted first followed by the elution of free Nsp1, as expected. If the molecular weight determined across the elution of the main peak in the complex is constant, the abundant species in the complexes are nearly monodispersed. The collected peak fractions were subjected to SDS-PAGE gel electrophoresis to confirm the formation of the complex.

Our SEC-MALS data revealed distinct retention times for both Fabs despite their similar molecular weights (Figure 3A,B). The SDS page gel inset, combined with MALS, showed similar-sized bands for both antibodies. Although 15497 and 15498 share a comparable molecular weight; however, 15498 displayed a longer retention time, likely due to some adherence to the column. This adherence resulted in a diverse molecular weight profile for 15498, suggesting the presence of both higher- and lower-weight molecules. On average, the MW determination leans toward a lower molecular weight owing to column adherence (Figure 3A,B). Furthermore, the higher standard deviation in the hydrodynamic radius of 15498 indicates size heterogeneity, correlating with a more extended structure for 15498 compared to 15497. However, the size of the complementarity-determining regions (CDRs) remains comparable in both the heavy and light chains of the tested antibodies. This extended structure of 15498 may permit it to reach and bind noncontiguous epitopes, unlike 15497, which primarily binds to linear/non-conformational epitopes. Overall, these synthetic antibodies, akin to the antigen, display the capability to adopt various conformations.

## 3. Discussion

Identifying the critical regulatory segment and understanding its effects on Nsp1 afford valuable insights into the functional and structural modulation of the protein. Antibody 15948 captures in binding to Nsp1 a higher energy state of the antigen much like a catalytic antibody [16] selectively stabilizes a high-energy transition state. This intriguing finding prompts the question of whether antibodies can leverage their binding energy, typically responsible for the catalytic ability of enzymes, to catalyze chemical transformations in their targets, going beyond their conventional role of merely serving as labeling molecules.

The Woods’ plot and supplementary kinetic plots (Appendix A) demonstrate increased HDX in most peptides spanning the secondary structure, excluding those at the binding site. In the context of 3D structure, the entire N-terminal domain experiences extensive conformational alterations, spanning regions both proximal and distal to the epitope, resulting in trapping a high-energy state.

This seems to be a novel observation as we were unable to find any precedents for antibody binding destabilizing nearly the entire structure of the antigen. Another possible explanation is the antibody binding destabilizes the native form of the antigen, but this seems less likely because bonding usually brings stability, not instability, to a molecule or to a complex.

Our previous temperature-dependent HDX study addressing the dynamics of full-length Nsp1 revealed that approximately one-third of the protein is unstructured, while the remaining two-thirds exhibits moderate compactness. Additionally, Nsp1 is not highly thermally stable, with a melting temperature of ~44 °C [15]. This suggests that the native structure of Nsp1 is readily modulated and that the energy barrier between the folded and unfolded states is not substantial, providing a small equilibrium population of the unfolded protein to interact with the antibody. Equilibrium causes more of the population to unfold and bind. The noncontiguous binding site spanning amino acid residues 20–27 and 73–85 can be viewed as a “hotspot” that becomes exposed and flexible in the higher energy state, and that hotspot binds to an antibody to stabilize the open state. Conversely, antibody 15497 exhibits the usual protection accompanying antigen/antibody binding.

These findings underscore the dynamic nature of the Nsp1 protein and its ability to undergo conformational changes. The HDX data show that, in principle, when antibodies interact with an antigen, the outcome can be stabilization or destabilization of the antigen structure [4]. A previous study demonstrated that dAbs may attach to the IBR of HOIP protein, promoting an alternative conformation that is more open compared to its standard solution-state structure, consequently leading to local destabilization or capturing a high-energy state [5]. In our study, we observed that the effect of 15498 is particularly pronounced, occurring over most of the structure of Nsp1.

Both HDX-MS and SEC-MALS unveil important characteristics of 15498 and its interactions with Nsp1. Whereas SEC-MALS provides valuable insights, it does not possess the requisite sensitivity to detect subtle conformational changes captured by HDX-MS. To validate the specific residues contributing to this conformational destabilization in Nsp1, we recommend conducting high-resolution studies such as X-ray crystallography and Cryo-EM in future investigations. HDX-based observations, however, stand as compelling and noteworthy in their own right.

The results also provide insight into antibody recognition and its impact on binding the high-order structure of an antigen. Further investigations into the functional implications of this hotspot can contribute to a better understanding of the immune response against Nsp1. Overall, the concept of developing an antibody that targets a transition state or a high-energy structure underscores the potential for conferring a catalytic effect on the corresponding chemical reaction. This exciting prospect paves the way for exploring the use of antibodies not only as catalysts but as probes of high-order structure, particularly for highly flexible proteins that serve many functions. The implications of such a discovery could offer new opportunities in immunology and biocatalysis, with applications in therapeutics and diagnostics. The possible application in therapeutics stems from the increase in entropy that is locked into the antibody/antigen complex and that may mitigate undesirable functions of a protein, especially for proteins that are multifunctional viral proteins.

## 4. Materials and Methods

### 4.1. Protein Purification

Nsp1 protein was purified according to recently published methods [13].

### 4.2. Fabs Selection and Characterization

The selection of Fab-phage clones, based on their binding to Nsp1, was conducted following previously described methods. Antibody variable domains of the selected clones, which exhibited binding to the Nsp1 antigen in Fab-phage ELISA, were sequenced by PCR amplification and the Sanger method, and unique clones were identified. The estimated affinities of the individual Nsp1 antigen-binding clones were determined as dissociation constants (*K_d_*) for 15497 and 15498 to be 0.9 nM and 1.4 nM, respectively, as measured by BLI [16]. The variable domain genes for clones exhibiting higher binding affinity (namely 15497 and 15498) were then cloned into vectors for expression.

### 4.3. IgG Expression and Purification

The selected variable domains were amplified by PCR and subcloned into pSCSTa-hIg1 and pSCST1-hk vectors. Expression constructs were then transfected into HEK293F cells using Fectopro (101000014, Polyplus, Illkirch-Graffenstaden, France), following the manufacturer’s instructions. The cells were incubated with shaking at 125 rpm at 37 °C for 4–5 days. After incubation, the cells were lysed, and the supernatant was collected after centrifugation. The lysate was applied to a Protein-A affinity column for purification. IgG proteins were eluted by using 100 mM glycine at pH 2.0 and neutralized with 2 M Tris at pH 7.5. The eluted proteins were then buffer exchanged into PBS at pH 7.4, concentrated, and analyzed by Western blotting. The IgGs were further characterized for their binding kinetics and immunofluorescence [16].

### 4.4. Hydrogen Deuterium Exchange Mass Spectrometry

HDX-MS was performed to measure the binding site and any structural perturbations of Nsp1 with antibodies 15497 and 15498. Both epitope and paratope mapping experiments were conducted as described elsewhere [12].

### 4.5. Epitope Mapping of Nsp1 with Fabs 15497 and 15498

Epitope mapping experiments used HDX-MS to investigate the differential behavior of Nsp1 in its bound and unbound states. In the unbound state, continuous HDX was conducted by diluting Nsp1 10-fold into a deuterated PBS buffer. The HDX was allowed to proceed for several time points (10, 30, 120, 300, and 3600 s) on ice. To quench the HDX, the deuterated samples were immediately mixed with 4 M guanidine-hydrochloride, 200 mM TCEP, pH 2.5. Fungal XIII solution (10 μL of 10 mg/mL) was pre-mixed with the quench solution, and enzymatic digestion was allowed to occur for 2 min. The samples were then flash-frozen in liquid nitrogen and stored at −80 °C until analysis via MS. Prior to injection into the LC/MS system, the frozen samples were thawed quickly. Online digestion was performed using a custom-packed pepsin column (2 × 20 mm). Simultaneous peptide trapping and desalting were carried out on a ZORBAX Eclipse XDB C8 column (2.1 mm × 15 mm) by using 0.1% formic acid aqueous solvent at a flow rate of 200 μL/min delivered by an HPLC pump. The valves, tubes, and analytical column were kept chilled, while the pepsin column was maintained at room temperature during the HDX measurements. Peptides were separated on a reversed-phase C18 column 2.1 mm × 50 mm, 2.5 µm X select-CSH) with the gradient; the organic solvent B (acetonitrile with 0.1% formic acid) was increased from 5% to 80% over 15 min. Following chromatographic separation, the peptides were analyzed with a Bruker Maxis HM Q-TOF mass spectrometer. The HDX data obtained were processed and analyzed using HDExaminer software (version 2.5.1, Sierra 433 Analytics, Inc., Modesto, CA, USA). The key parameters considered during data analysis by HDExaminer included (1) excluding deuterations in the first two residues, where back exchange is particularly rapid, (2) ensuring the elution time window of the peptides remains within 0.50 min of the non-deuterated window, (3) recognizing that proline does not undergo exchange, and (4) noting the absence of an assessment of back exchange corrections in the analysis (back exchange considerations are less important for differential measurements than for single determinations). HDExaminer computes the centroid of an isotope cluster using experimental isotope distribution (A, A + 1, A + 2, … where A is the monoisotopic mass). When set to “Theoretical isotope clusters”, HDExaminer 2.5.1 determines the centroid of the isotope pattern, including the deuteriums, accurately aligns it with the experimental data, and outputs the extent of HDX.

For the bound state, Nsp1 and antibody 15497 were mixed in a molar ratio of 1:2 on ice and incubated for 30 min before initiating the HDX process. The experimental setup and molar ratios were kept consistent for the second antibody, 15498, for its epitope mapping experiments. All HDX experiments were conducted in duplicate. The antibody stock was initially prepared in PBS buffer. Quench solution and deuterated buffers were then pre-prepared in bulk, frozen in liquid N2, and stored at −80 °C. Upon thawing, these solutions were promptly utilized for experiments. Different time points for both bound and unbound states were collected on the same day, flash-frozen in liquid N2, and stored at −80 °C. Subsequently, samples from both states were processed and analyzed via mass spectrometry on the same day. Prior to initiating the HDX reactions, the pH of the quench solution and deuterated PBS buffers were re-checked for consistency. Nsp1 samples were subjected to one freeze–thaw cycle for bound and unbound state HDX measurements.

### 4.6. Paratope Mapping of 15497 and 15498 with Nsp1

The HDX experimental setup for paratope mapping was the same, except the concentration of TCEP in the quenching solution was higher (4 M guanidine-hydrochloride, 500 mM TCEP, pH 2.5). The antigen was added in excess to the antibody at a ratio of 1:2.

### 4.7. Peptide Mapping and HDX Data Analysis

Before initiating epitope mapping, peptide mapping of Nsp1 was performed. A stock solution of 50 μM Nsp1 was prepared in PBS (pH 7.4) buffer and used for peptide mapping in the LC/MS/MS mode. The cycle time was 3 s in the Bruker Maxis HM Q-TOF mass spectrometer. The MS/MS data were analyzed utilizing Byonic and Byologic (Protein Metrics, San Carlos, CA, USA).

To ensure accuracy and minimize false positive results, a reversed protein sequence control was implemented during peptide identification. Statistical analysis was performed to evaluate the differences in deuterium uptake between the bound state (Nsp1 + antibodies) and the unbound state of Nsp1. Cumulative differences in hydrogen–deuterium exchange (HDX) between the two states were calculated, and those exceeding the significance limit were considered statistically significant. The propagated error for each peptide was determined as the square root of the sum of squared standard deviation values across all time points for both states. Differences greater than the propagated error were considered statistically significant [15].

## Figures and Tables

**Figure 1 ijms-24-17342-f001:**
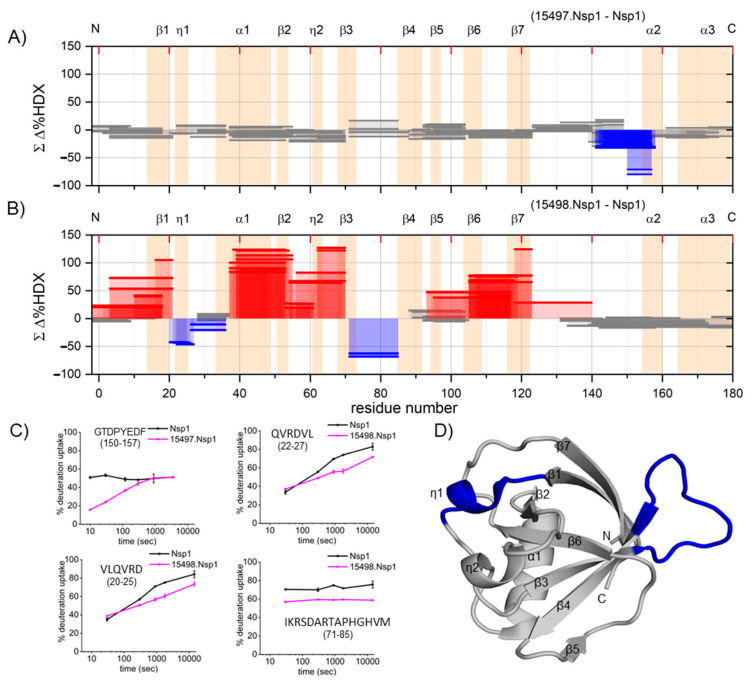
Epitope mapping by HDX-MS: The cumulative HDX differences in the Nsp1-bound antibody and the free Nsp1; (**A**) 15497 and (**B**) 15498, are shown in the Woods’ plot format. Secondary structural elements are labeled on top of each plot (peach shading); η1 and η2 represent two short 3_10_ helices. Peptides with significant protection in blue and significant exposure in red are also shown. Peptides derived from unaffected regions are shown in gray. (**C**) Kinetic plots for selected peptides representing the potential binding sites on free Nsp1 to give 15497 or 15498 bound complexes. (**D**) Regions showing the potential binding site for 15498 are highlighted in blue on the Nsp1 N-terminal domain structure (PDB ID 7K7P). The secondary structural elements are labeled on the three-dimensional structure.

**Figure 2 ijms-24-17342-f002:**
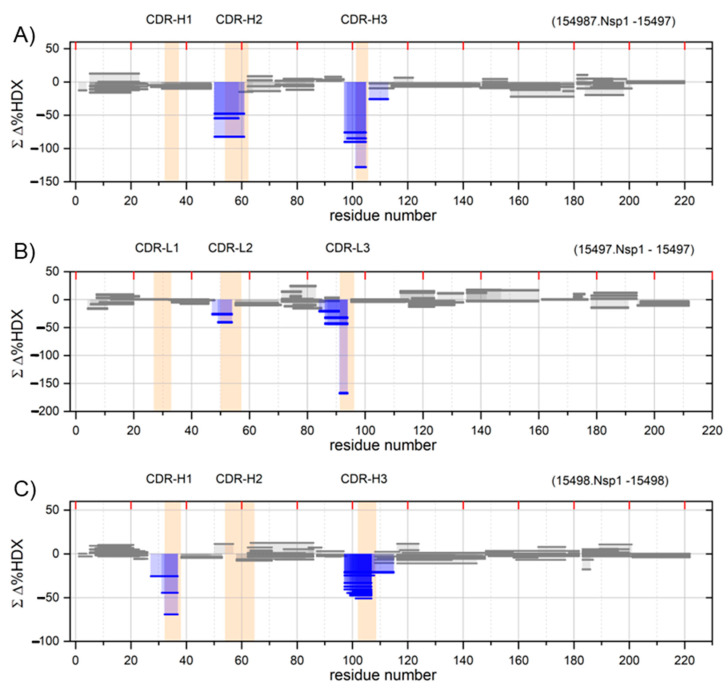
Paratope mapping by HDX-MS. Woods’ plot shows the cumulative HDX differences of the Nsp-bound antibody minus the free antibody. (**A**) 15497-Heavy chain, (**B**) 15497-Light chain, (**C**) 15498-Heavy chain, (**D**) 15498-Light chain. The complementary determining regions are labeled on top of the plot and depicted with peach shading. The blue color peptides (regions) indicate significant protection (paratopes), and grey color peptides show the unaffected segments post antibody binding.

**Figure 3 ijms-24-17342-f003:**
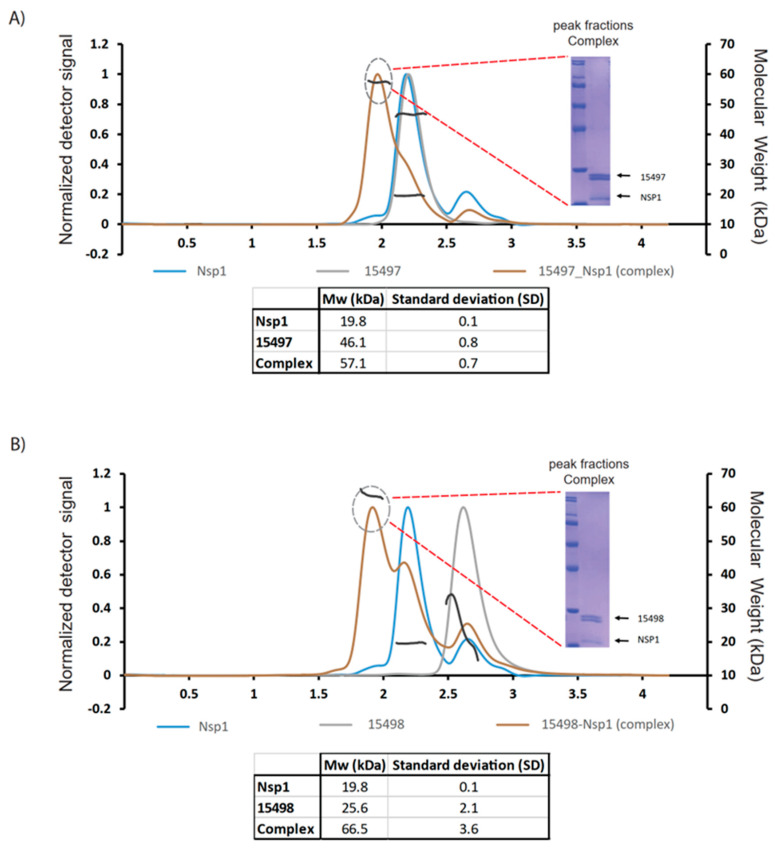
SEC-MALS of antibody with Nsp1. (**A**) The chromatographic profiles illustrate Fab 15497 and its complex with Nsp1, displaying their respective molar masses in relation to their peaks. In the inset, the SDS-PAGE gel illustrates the peak fractions for the Fab-Nsp1 complex. (**B**) Chromatographic profiles show Fab 15498 and its complex, with their respective molar masses indicated by the peaks. SDS-PAGE gels further confirm peak fractions of the Fab-Nsp1 complex. These experiments were conducted in triplicate. The Tables in the Figure show the molecular weight of free Nsp1, Fabs, and complex with their corresponding standard deviations.

## Data Availability

All data gathered for this project are reported in the main manuscript and the Appendix A.

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
