# Peer review of "Antibody Binding Captures High Energy State of an Antigen: The Case of Nsp1 SARS-CoV-2 as Revealed by Hydrogen–Deuterium Exchange Mass Spectrometry"

_ijms, 2023, doi:10.3390/ijms242417342_

Round 1

Reviewer 1 Report

Comments and Suggestions for Authors

ijms-2736266

Antibody Binding Captures High Energy State of an Antigen: The Case of Nsp1 SARS-CoV-2 as Revealed by Hydrogen-Deuterium Exchange Mass Spectrometry

The manuscript by Kant et al described the characterization of the binding of two antibodies (Fabs 15497 and 15498) with Nsp1 by using hydrogen-deuterium exchange mass spectrometry (HDX-MS). The authors have appropriately presented the data and structured the manuscript. The data are sufficient for the conclusion. I suggest some minor comments to improve this manuscript as follows.

1. Introduction: The authors should include a short introduction of the HDX-MS method and clarify the necessity of using HDX-MS to study the binding of two antibodies (Fabs 15497 and 15498) with Nsp1.

2. The authors should include software parameters to process and analyze HDX data.

Comments on the Quality of English Language

Minor editing of English language required

Author Response

Does the introduction provide sufficient background and include all relevant references?

(x)

( )

( )

( )

Are all the cited references relevant to the research?

( )

(x)

( )

( )

Is the research design appropriate?

(x)

( )

( )

( )

Are the methods adequately described?

(x)

( )

( )

( )

Are the results clearly presented?

(x)

( )

( )

( )

Are the conclusions supported by the results?

(x)

( )

( )

( )

Comments and Suggestions for Authors

ijms-2736266

Antibody Binding Captures High Energy State of an Antigen: The Case of Nsp1 SARS-CoV-2 as Revealed by Hydrogen-Deuterium Exchange Mass Spectrometry

The manuscript by Kant et al described the characterization of the binding of two antibodies (Fabs 15497 and 15498) with Nsp1 by using hydrogen-deuterium exchange mass spectrometry (HDX-MS). The authors have appropriately presented the data and structured the manuscript. The data are sufficient for the conclusion. I suggest some minor comments to improve this manuscript as follows.

  1. Introduction: The authors should include a short introduction of the HDX-MS method and clarify the necessity of using HDX-MS to study the binding of two antibodies (Fabs 15497 and 15498) with Nsp1.

As per the reviewer suggestion, we added these new sentences explaining the HDX-MS methodology in the introduction , please see the page 2 and lines54-82 for the details.

The standard bottom-up HDX-MS process involves these steps: (1) incubation with D2O (labeling), (2) quenching of the HDX by adding acidified denaturant, (3) digestion of the Nsp1 and/or antibody with acid-stable proteases, (4) chromatographic separation of peptides followed by mass spectrometric detection, and (5) semiautomatic data analysis. Proteins in solution are mixed into a D2O-based buffer in the first step, allowing labile hydrogens to exchange with deuterium from the solvent. Both backbone (N−H) and some side-chain hydrogens exchange, but HDX of protein side chains is not measured owing to their rapid exchange in and out following the quench. Amide hydrogens in the backbond, however, exchange over slowly (minutes to hours), allowing them to be detected with mass spectrometry. HDX is catalyzed by both acid and base, giving rise to a minimum in exchange rate occurring at pH around 2.5, motivating the choice of an acid quench. Using a combination of proteases ensures the generation of small to medium-sized peptides over most of the protein, offering detailed, high spatial resolution. Semi-automated data analysis software calculates the mass shift due to deuterium incorporation. The software examines the isotopic distribution of a peptide or protein in its undeuterated state and establishes  its centroid. Comparing this with the shifted centroid at later time points reveals the most probable extent of HDX. The results of the analysis is pictured as a kinetics graph plotting deuterium uptake (Y-axis) against time (X-axis). In most HDX experiments, researchers conduct differential footprinting, comparing two states (e.g., free versus ligand-bound or wild type versus mutant). Deuterium differences are computed by subtracting HDX values between these states, uncovering both local and allosteric impacts of the binding.

Our previous HDX findings regarding Nsp1 show that approximately one-third of the protein lacks a compact, well-defined structure, consistent with its flexibility and posing challenges for crystallizing the full-length Nsp1. This observation aligns with the crystal structure of Nsp1, which had to be taken of a truncated version spanning residues from 10 to 127. Notably, we chose HDX for this study because HDX-MS is rapid, solution-based, is sensitive to protein dynamics and flexibility, and doesn't depend on protein size or complexity.

  1. The authors should include software parameters to process and analyze HDX data.

We added information related to the parameters used in HDExaminer, used for processing the HDX data, in the method section. Please look into the page 9 and line number, 322 for the extra added information.

“The key parameters considered during data analysis by HDExaminer included (1) excluding deuterations in the first two residues, where back exchange is particularly rapid, (2) ensuring the elution time window of the peptides  remains within 0.50 minutes of the non-deuterated window, (3) recognizing that proline does not undergo exchange, and (4) noting the absence of an assessment of back exchange corrections in the analysis (back exchange considerations are less important for differential measurements than for single determinations). HDExaminer computes the centroid of an isotope cluster using experimental isotope distribution (A, A+1, A+2,… where A is the monoisotopic mass. When set to "Theoretical isotope clusters," HDExaminer determines the centroid of the isotope pattern including the deuteriums, accurately aligns it with the experimental data, and outputs the extent of HDX.

Comments on the Quality of English Language

Minor editing of English language required

Submission Date

08 November 2023

Date of this review

20 Nov 2023 09:14:43

Reviewer 2 Report

Comments and Suggestions for Authors

The authors have presented their study on the effects of antibody binding to the Nsp1 protein of SARS-CoV-2. Utilizing HDX-MS experiments with two distinct antibodies, they propose that one antibody's binding did not alter Nsp1's dynamics, while the other's made Nsp1 more flexible than its unbound state. This finding contrasts with the common observation that antibody binding typically stabilizes or rigidifies an antigen. Although the results are intriguing, the discussion is insufficient, and the manuscript seems preliminary. I would like to offer several questions and comments regarding their findings.

First, is Nsp1 inherently structurally flexible? The Introduction suggests that Nsp1's multifunctionality necessitates such flexibility for effective interactions with various viral and host factors. Yet, the main text lacks evidence to substantiate this claim. The presence of an Nsp1 crystal structure implies some degree of structural rigidity, as crystallization usually requires it. Figure 1 implies that the binding of antibody 15498 to Nsp1 induces flexibility in certain regions, seemingly around the identified epitopes. Consequently, the paper's title may be misleading, as there's no direct evidence that arecognized state of Nsp1 by the antibody is a "high energy" state, akin to catalytic antibodies. It seems rather that antibody binding results in this "high energy" state. The authors themselves seem to note this in the subtitle of section 2.2: “Antibody binding causes widespread conformational change.” Hence, the sequence of events does not seem to align with the authors' argument.

The SEC-MALS results indicate that antibody 15948 has a more extended structure than 15497. What could account for this? Given that the molecular weights suggest 15948 is considerably smaller than 15497 (25.6 kDa vs 46.1 kDa), could shorter CDRs be a factor? Then, it raises the question of how such a smaller antibody can adopt an extended structure to recognize spatially distant regions on Nsp1 (as shown in Figure 1D). Interpreting HDX-MS data can indeed be complex, but a more thorough discussion would be valuable. Molecular dynamics simulations or molecular modeling of the antibody-antigen complexes, informed by HDX-MS data, could substantiate their findings.

Including binding affinity data for the two antibodies towards Nsp1 and discussing the relationship between binding affinity and epitope flexibility would be enlightening.

In Figure 1, secondary structures are denoted with Greek symbols. While alpha and beta are clear, the meaning of eta is not. Additionally, in Figure 1D, it would be helpful if the epitope of 15497 were also highlighted. The label of alpha 2, identified as an epitope of 15497 in Figure 1A, is not present.

Author Response

Yes

Can be improved

Must be improved

Not applicable

Does the introduction provide sufficient background and include all relevant references?

( )

(x)

( )

( )

Are all the cited references relevant to the research?

( )

(x)

( )

( )

Is the research design appropriate?

( )

( )

(x)

( )

Are the methods adequately described?

(x)

( )

( )

( )

Are the results clearly presented?

( )

( )

(x)

( )

Are the conclusions supported by the results?

( )

( )

(x)

( )

Comments and Suggestions for Authors

The authors have presented their study on the effects of antibody binding to the Nsp1 protein of SARS-CoV-2. Utilizing HDX-MS experiments with two distinct antibodies, they propose that one antibody's binding did not alter Nsp1's dynamics, while the other's made Nsp1 more flexible than its unbound state. This finding contrasts with the common observation that antibody binding typically stabilizes or rigidifies an antigen. Although the results are intriguing, the discussion is insufficient, and the manuscript seems preliminary. I would like to offer several questions and comments regarding their findings.

First, is Nsp1 inherently structurally flexible? The Introduction suggests that Nsp1's multifunctionality necessitates such flexibility for effective interactions with various viral and host factors. Yet, the main text lacks evidence to substantiate this claim. The presence of an Nsp1 crystal structure implies some degree of structural rigidity, as crystallization usually requires it. Figure 1 implies that the binding of antibody 15498 to Nsp1 induces flexibility in certain regions, seemingly around the identified epitopes. Consequently, the paper's title may be misleading, as there's no direct evidence that arecognized state of Nsp1 by the antibody is a "high energy" state, akin to catalytic antibodies. It seems rather that antibody binding results in this "high energy" state. The authors themselves seem to note this in the subtitle of section 2.2: “Antibody binding causes widespread conformational change.” Hence, the sequence of events does not seem to align with the authors' argument.

On the matter of capturing a high energy state, we recognize the possibility of antibody binding destabilizes the antigen in the paragraph in sect 2.2: “This seems to be a novel observation as we were unable to find any precedents for antibody binding destabilizing nearly the entire structure of the antigen. Another possible explanation is the antibody binding destabilizes the native form of the antigen, but this seems less likely because bonding usually brings stability, not instability, to a molecule or to a complex.”  We changed the misleading subtitle for this section.”

We appreciate the reviewer's acknowledgment of the phrase "inherently structurally flexible." Our earlier temperature-dependent HDX study (Mishra et al., 2023) specifically highlighted Nsp1's flexibility. Our research emphasizes that the N-terminus, linker, and C-terminus, constituting nearly one-third of the total residues, lack defined structures and are absent in the Nsp1 crystal structure. Additionally, various loops and connecting secondary structures display dynamic properties. These collective observations support our assertion that Nsp1 is inherent flexible, which is consistent with its role in viral infection. Importantly, our use of the term "inherently structurally flexible" doesn't imply a complete lack of secondary structure within the protein. We added a condensed version of this reply in the text on page X, lines yy

We apologize for any confusion caused by our earlier explanation about the flexibility shift. We have added a statement in the discussion section, page 8 and line no. 229 for more clarification.

The Woods plot and supplementary kinetic plots demonstrate increased HDX in most peptides spanning the secondary structure, excluding those at the binding site. In the context of the 3D structure, the entire N-terminal domain experiences extensive conformational alterations, spanning regions both proximal and distal to the epitope, resulting in entrapment within a high-energy state.

The SEC-MALS results indicate that antibody 15948 has a more extended structure than 15497. What could account for this? Given that the molecular weights suggest 15948 is considerably smaller than 15497 (25.6 kDa vs 46.1 kDa), could shorter CDRs be a factor? Then, it raises the question of how such a smaller antibody can adopt an extended structure to recognize spatially distant regions on Nsp1 (as shown in Figure 1D).

We added the revised text in the result section to clarify this concern, page 6 and lines 194-209

 "Our SEC-MALS data revealed distinct retention times for both Fabs despite their similar molecular weights. The SDS page gel inset, combined with MALS, showed similar-sized bands for both antibodies. However, although 15497 and 15498 share a comparable molecular weight, 15498 displayed a longer retention time, likely due to its adherence to the column. This adherence resulted in a diverse molecular weight profile for 15498, indicating the presence of both higher and lower weight molecules. On average, it leans towards a lower molecular weight due to this column adherence. Furthermore, the higher standard deviation in the hydrodynamic radius of 15498 indicates size heterogeneity, correlating with a more extended structure for 15498 compared to 15497. However, the size of the complementarity-determining regions (CDRs) remains comparable in both the heavy and light chains of the tested antibodies. Overall, these synthetic antibodies, akin to the antigen, display the capability to adopt various conformations.

Interpreting HDX-MS data can indeed be complex, but a more thorough discussion would be valuable. Molecular dynamics simulations or molecular modeling of the antibody-antigen complexes, informed by HDX-MS data, could substantiate their findings.

We value the reviewers' suggestion, acknowledging its potential relevance for future studies. Integrating crystallographic studies with Molecular Dynamics simulations or molecular modeling could significantly enhance our understanding. Some parts of the protein were truncated to obtain a crystal structure, and given the flexibility of the antigen, it is difficult to input a structure to the docking computation Furthermore, the primary objective of our current study is to identify binding sites and elucidate remote conformational changes at the peptide level. Thus, we view this study as a foundational step for subsequent investigations focused on exploring residue-level details.

Including binding affinity data for the two antibodies towards Nsp1 and discussing the relationship between binding affinity and epitope flexibility would be enlightening.

We reported the dissociation constants 15498 in the method section as 1.4 nM determined via BLI. These values indicate a notably robust binding affinity for both antibodies.

 We also added a sentence in the results section, page 4, lines 154-157.

It's intriguing that the kinetic plots depicting Nsp1 interactions, with and without antibodies, revealed that despite global delocalization, 15497 maintained its binding to Nsp1, indicating a strong and resilient interaction.

In Figure 1, secondary structures are denoted with Greek symbols. While alpha and beta are clear, the meaning of eta is not.

We added additional text in the figure 1, page 4, line 143 “captionη1 and η2 represent two short 310 helices”.

Additionally, in Figure 1D, it would be helpful if the epitope of 15497 were also highlighted. The label of alpha 2, identified as an epitope of 15497 in Figure 1A, is not present.

A sentence was added to address the concern above, see page 3, lines 113-115, Regrettably, due to the absence of the C-terminus in the crystal structure (that part of the protein was truncated), we are unable to depict the epitope of 15497 on the crystal structure.

Submission Date

08 November 2023

Date of this review

17 Nov 2023 05:32:01

Round 2

Reviewer 2 Report

Comments and Suggestions for Authors

The authors addressed all of my previous comments properly.